# Difference Between Walking Parameters During 6 Min Walk Test Before and After Abdominal Surgery in Colorectal Cancer Patients

**DOI:** 10.3390/cancers17111782

**Published:** 2025-05-26

**Authors:** Nikolina Santek, Sanja Langer, Iva Kirac, Danko Velemir Vrdoljak, Gordan Tometic, Goran Musteric, Ljiljana Mayer, Maja Cigrovski Berkovic

**Affiliations:** 1Department of Rheumatology, Physical Medicine and Rehabilitation, Sestre Milosrdnice UHC, 10000 Zagreb, Croatia; 2Faculty of Kinesiology, University of Zagreb, 10000 Zagreb, Croatia; 3Department of Chemistry, Sestre Milosrdnice UHC, 10000 Zagreb, Croatia; 4Department for Tumors, Sestre Milosrdnice UHC, 10000 Zagreb, Croatia

**Keywords:** colorectal cancer, major abdominal surgery, 6 min walk test

## Abstract

This study aims to show the difference in 6 min walk test parameters before and after major abdominal surgery in patients with colorectal cancer. We measured walking speed, number of steps, distance, and heart rate during walking. We analyzed this parameter overall and by groups. Grouping variables were gender, age, oncological diagnosis, other comorbidities and drug use, neoadjuvant therapy before surgery, level of physical activity before surgery, BMI, and duration of surgery. Based on our results, we can predict cardiorespiratory response after surgery during walking. This is primarily applied to overweight and obese patients whose heart rate during walking is significantly higher after surgery.

## 1. Introduction

Colorectal cancer (CRC) is the third most commonly diagnosed cancer and the fourth most common cause of cancer-related death in both men and women worldwide. At present, CRC accounts for 10% of all malignancies in women and 12% in men, ranking second and third in terms of incidence, respectively [1]. Also, it is a leading cause of death in men younger than 50 years [2]. More than half of all CRC-related causes and deaths are attributable to modifiable risk factors such as unhealthy diet, physical inactivity, smoking, high alcohol consumption, and excess body weight [3]. Surgery for colon cancer represents the only curative option and can favorably impact even the metastatic disease, so it is included in most treatment concepts [4,5,6,7]. The main goals of surgical treatment for CRC are avoiding recurrence and metastatic spread, treatment of complications, and maintaining quality of life [8].

While the in-hospital mortality following CRC surgery in modern-day practice is low, patients still suffer from significant procedure-specific postoperative morbidities [9]. Adverse events following major abdominal surgery are often linked to the severity of pre-existing comorbidities and the functional ability of patients to meet the additional metabolic demands required during complex surgery [10]. Approximately one-third of patients who undergo CRC resection experience postoperative complications, which can delay recovery, prolong hospitalization, and cause unplanned hospital readmissions, while chronic illnesses and their severity additionally impair short- and long-term physical functioning and health-related quality of life [11,12].

CRC surgery has experienced some significant preoperative improvements over the last several decades. Preoperative risk assessment of nutrition, frailty, and sarcopenia, followed by intervention to optimize the patient’s outcomes or an adapted surgical strategy have improved the postoperative course.

Rapid advancements in surgical and perioperative care have significantly shortened hospital length of stay [13]. From a patient’s perspective, recovery includes the return to a preoperative level of independence and well-being [14]. One recovery outcome after surgery is physical function, which likely influences other recovery domains.

Cardiopulmonary exercise testing (CPET) is now performed widely before surgery [15,16] and is considered to be the most objective and precise means of evaluating pre-surgical physical fitness [17,18,19]. The six-minute walk test (6MWT) is a relatively quick, simple, cheap, safe, and clinically acceptable measure of assessing functional exercise capacity. It has been shown to correlate well with cardiorespiratory exercise testing in patients with cardiopulmonary disease or undergoing major non-cardiac surgery [20]. Also, it is a prognostic factor [21]. Evidence regarding the ability of the 6MWT to predict patient outcomes in perioperative settings is limited to cohort studies with a sample size of approximately 100 participants. In patients with major abdominal or thoracic surgery, the 6MWT distance was predictive of postoperative complications [20,22] and duration of hospital stay [23]. In some clinical situations, the 6MWT provides information that may be a better index of a patient’s ability to perform daily activities than peak oxygen uptake; for example, 6MWT distance correlates better with formal quality of life measures [24]. Also, the 6MWT is a practical and validated tool in research studies for cancer patients, including CRC patients [25,26,27].

This study was primarily designed to investigate whether parameters of walking performance, when combined with other preoperatively recorded variables, were associated with postoperative walking performance in a mixed cohort of patients scheduled for major abdominal cancer surgery. The secondary aim was to determine the impact of quality of life before surgery, quality of recovery after surgery, and laboratory data on walking performance before and after surgery.

## 2. Materials and Methods

### 2.1. Study Design

This manuscript reports a prospective cohort study evaluating different parameters during walking before and after surgery between March 2022 and September 2023. The Ethical Committee of Sestre Milosrdnice University Hospital Centre in Zagreb, Croatia, approved the study conducted in accordance with good clinical practice and the Declaration of Helsinki (EP-9941/19-13).

### 2.2. Participants

CRC patients in all stages of cancer were eligible for the trial if they were scheduled for elective open or laparoscopic abdominal cancer surgery under general anesthesia in the Department of Surgical Oncology, Department for Tumors, Sestre Milosrdnice University Hospital Centre. All participants were Croatian speakers, and all patients gave their written consent for participation.

Excluded were patients with cardiorespiratory (uncontrolled hypertension, unstable angina pectoris, uncontrolled cardiac arrhythmia, or aortic aneurism more than 6.5 cm), musculoskeletal, and/or neurologic (impaired mobility, orthopedic surgeries where recovery is not complete, lower limb amputations, degenerative or inflammatory diseases of the locomotor system that result in reduced mobility and difficulty walking (osteoarthritis, rheumatoid arthritis, ankylosing spondylitis, etc.), anemia where hemoglobin was below 80, cachexia, and conditions after a cerebrovascular accident where movement and walking are complex) conditions, as well as those needing revision surgery or with a second surgery within 6 months, those with psychiatric conditions or cognitive disorders, frail patients.

### 2.3. Study Protocol

As per standard care, all participants listed for CRC surgery were evaluated by the hospital’s multidisciplinary team (surgeon, anesthesiologist, and nurse). They received nutritional support pre- or post-operatively, where needed, provided by a nurse/stomal therapist. Also, all participants were mobilized the day after surgery.

We included patients in the study after hospitalization, which is 2 days before surgery. We performed the postoperative measurement 7 days after surgery, approximately 1–3 days before discharge from the hospital (Figure 1).

After admission to the hospital, participants who signed an informed consent form were included in a prospective cohort study. They completed a short questionnaire about health and physical status and preoperative physical activities.

After a short questionnaire, the subjects completed the EORTC QLQ-C30 quality of life questionnaire (European Organization for Research and Treatment of Cancer). We used the Quality of Recovery Questionnaire (QoR 15) 7 days after surgery.

Quality of life (QOL) was assessed using the Croatian version of the EORTC QLQ-C30 core questionnaire, version 3 [28]. The European Organization of Cancer Research and Treatment developed this questionnaire. The QLQ-C30 consists of 30 items, including five functional scales (about global health, physical, role, cognitive, emotional, and social function), three symptoms scales (fatigue, nausea/vomiting, pain), and six individual items (symptoms) usually associated with malignant disease: dyspnea, insomnia, appetite, constipation, diarrhea, and financial difficulties. The scales and items were evaluated on a Likert scale of 4 levels, ranging from 1 (not at all) to 4 (almost always). A higher number of points correlated to poorer functioning and more symptoms. The global health/quality of life scale consisted of a seven-point linear analog scale where a higher score indicated greater satisfaction with the global health status and quality of life [29]. Patients’ responses were combined and computed on a 0 to 100 scale according to the scoring manual provided by EORTC. It is a self-reported and validated questionnaire that is valid and reliable in a multicultural setting.

The QoR 15 (quality of recovery 15) questionnaire is a validated tool to assess postoperative recovery [30]. It ranges from 0 to 150 points, with a higher score indicating a better quality of recovery. Fifteen questions assess five domains of patient-reported health status: physical comfort, independence, pain, psychological support, and emotional status. The 11-point numerical rating scale leads to a minimum score of 0 (extremely poor recovery) and a maximum score of 150 (excellent recovery) [31]. The QoR 15 is the most widely reported measure of patient-assessed QoR after surgery [32,33].

Patients performed a six-minute walk test (6MWT) two days before surgery and seven days after surgery. The 6MWT is a tool for measuring the functional status of fitness. During the test, participants walked at their normal pace for six minutes.

Absolute contraindications for the 6MWT include unstable angina and myocardial infarction during the previous month, while relative contraindications include a resting heart rate of more than 120 bpm, a systolic blood pressure higher than 180 mm Hg, and a diastolic blood pressure higher than 100 mm Hg.

Participants should rest for 15 min before the beginning of the 6MWT. Before starting the test, blood pressure, heart rate, and oxygen saturation were measured. The same measurements were repeated after the 6MWT.

For the 6MWT, we used a 60 m long hospital corridor. Each subject was given clear instructions on how to perform the test. The aim of the test was to walk as far as possible. A physiotherapist accompanied the subject throughout the test and noted how much time had passed every minute. The subject was also instructed that it was not advisable to talk during the test. If there were any questions or uncertainties, they should ask them before or after the test unless they had another problem with the 6MWT or the examiner asked them a question. They must say if they felt chest pain or dizziness during the test.

The 6MWT was conducted with a Garmin Vivosmart 4 activity tracker (Garmin Ltd., Olathe, KS, USA). During the 6MWT, we measured average and maximal heart rate, walking distance, number of steps, and speed. Patients performed the 6MWT at the same time of day before and after surgery.

The laboratory data extracted were from the day before surgery, 24 h after surgery, and 7 days after surgery (erythrocytes, hemoglobin, leukocytes, and CRP). For this study, we also used some laboratory findings: hemoglobin and erythrocytes, which carry oxygen through the blood to muscle cells. Adequate oxygenation of muscle cells is especially important during physical activity, such as walking. Additionally, we took inflammatory parameters that are routinely available to us (leukocytes and CRP) to determine whether inflammatory affect reduced physical capacity after surgery. 

### 2.4. Outcome Measurement

The primary endpoint was defined as the difference in walking performance (walking distance, number of steps, average and maximal speed, and average and maximal heart rate) before and after major abdominal surgery for CRC patients. We also analyzed the differences in walking performance between the different groups of participants (according to gender, age, another diagnosis, therapy before surgery, physical activity before surgery, body mass index, type of incision, duration of surgery, and pathohistological analysis).

Secondary endpoints were quality of life before surgery, quality of recovery before surgery, and its impact on walking performance. Also, we used data from laboratory blood analysis and their impact on average and maximum heart rate during walking.

### 2.5. Statistical Analysis

Data were analyzed in the Statistica package, version 14.0.0 (TIBCO Data Science). Patient demographics, diagnosis, and treatment characteristics were presented as categorical variables using a frequency table. Continuous variables were presented with mean (standard deviation SD) or median (interquartile range IQR). For continuous variables, an independent samples *t*-test or the Mann–Whitney U test, as appropriate, was performed to analyze differences between the two measurements.

For the analysis of walking performance (number of steps, walking distance, average and maximal speed, average and maximal heart rate) during 6MWT, we used ANOVA. For this test, specific adjustments to the data were necessary. It was essential to divide the subjects into groups according to age, gender, diagnosis, other diagnoses, cardiovascular drug used, weight loss, digestive symptoms, treatment after previous surgery, physical activity before surgery, BMI, type of surgery, and duration of surgery. We compared variables between the groups with ANOVA for dependent measurements. We used Tukey for data correction. A multiple logistic regression analysis was performed to identify independent predictors of heart rate parameters during 6MWT using laboratory data, a quality of life questionnaire, and a quality of recovery questionnaire. The significance threshold was set at *p* value < 0.05.

## 3. Results

A total of 84 patients were assessed for eligibility from 18 March 2022 to 1 October 2023. Three patients were excluded from the second measurement due to infection. Three patients had revision surgery due to anastomose dehiscence; one patient had revision surgery due to bleeding; one patient could not perform a second measurement due to general weakness; and four patients refused secondary measurement after surgery for personal reasons. In the final analysis, we included 72 patients with a mean age of 62.48 years (range 39–81, SD = 10.03). The basic characteristics of the patients are shown in Table 1. We present the results of the quality of life questionnaire (EORTC QLQ C-30) taken 2 days before surgery and the quality of recovery questionnaire (QoR 12) taken 7 days after surgery in Table 2.

We compared the number of steps, walk distance, average and maximal walk speed, and average and maximal heart rate before and after surgery. There was a significant difference between maximal heart rate before and after surgery: 96 bpm and 107 bpm, respectively, *p* = 0.000. The average heart rate was also significantly different: 65 bpm before surgery and 99 after surgery, *p* = 0.008. There were no significant differences in maximal speed *p* = 0.844 (5.47 km/h before surgery and 5.74 km/h after surgery). Still, there was a significant difference between average speed: *p* = 0.020 (4.32 km/h before and 4.22 km/h after surgery). A significant difference was found in the number of steps *p* = 0.000 (581.5 before surgery and 544 after surgery) and walk distance *p* = 0.088 (354.2 before surgery and 214.5 after surgery). The measured values during the 6MWT are presented in Table 3.

### Differences Between Groups

We compared measured values during the 6MWT between the groups. Grouping variables used included gender (men and women), age (younger than 60, between 60 and 80, and older than 80 years), localization of tumor (rectum, sigmoid colon, right hemicolon, left hemicolon, colorectal), other diseases (arterial hypertension, heart disease, type 2 diabetes, other oncological diseases), cardiovascular drugs (beta blockers ACE inhibitors, angiotensin II receptor blockers), treatment before surgery (chemotherapy/radiotherapy, chemotherapy or radiotherapy) level of physical activity before surgery (low, medium, or high level), BMI (normal weight, overweight, obese), type of surgery (major open abdominal or laparoscopic), and duration of surgery (up to 2 h, between 2 and 3 h, more than 3 h).

Our data show a statistically significant difference between men and women in the distance walked. Men walked a greater distance before and after surgery (*F* = 4.99, *p* = 0.02). There is also a statistically significant difference in the number of steps according to patients’ age. After the surgery, those over 75 took significantly fewer steps (*F* = 2.90, *p* = 0.02). Also, subjects with arterial hypertension took significantly fewer steps after surgery (*F* = 4.95, *p* = 0.01). A statistically significant difference was detected in the average and maximum heart rates during walking when comparing body mass index (average heart rate *F* = 5.72, *p* = 0.00, maximum heart rate *F* = 2.52, *p* = 0.04). Patients living with obesity had a significantly higher heart rates during walking before and after surgery than patients with normal body weight. Participants who were overweight had comparable heart rates to patients with normal body weight during walking before surgery. Still, their increased measurements after surgery were significantly higher than those seen in people with normal body weight. When we compared the groups by cardiovascular drugs, we had a statistically significant difference in the group who used beta-blockers: *F* = 3.23, *p* = 0.04 in average heart rate. Participants who used beta-blockers had lower heart rates before and after surgery and lower differences between these two values.

Also, data show that there are statistically significant differences in number of steps (*F* = 4.87, *p* = 0.01), average speed during walking (*F* = 3.55, *p* = 0.03), maximal speed (*F* = 3.86, *p* = 0.02), and average heart rate during walking (*F* = 4.12, *p* = 0.02) between patients with DMT2 and without DMT2. Patients with DMT2 took a significantly lower number of steps than patients without DMT2 after surgery. Also, patients with DMT2 had lower average and maximal speeds during walking after surgery. Those patients had higher average heart rates after surgery. We did not find any statistically significant difference in the regression model.

## 4. Discussion

Major surgery leads to higher metabolic demands, which are matched in healthy individuals by concomitant cardiac output and ventilation increases. We performed a prospective comparison study to compare walk distance, number of steps, speed, and heart rate during 6MWT before and after CRC surgery for different subgroups of patients.

The data presented in this study show that postoperative patients had a higher maximum and average heart rate during the 6MWT. The average heart rate during walking was higher in overweight and obese people. It should be noted that people living with obesity had significantly higher heart rates before surgery. The situation was similar with the maximum heart rate during the 6MWT. It is important to note here that people with arterial hypertension and ischemic heart diseases did not show a significant difference in average and maximum heart rate during walking before and after surgery. The reason may be the use of beta-blockers in pharmacologic therapy (all 16 participants with beta-blocker therapy were in the arterial hypertension group). Postoperatively, the average and maximum walking speed was lower, especially in patients with DMT2. All results are shown in Figure 2a–d.

It is also important to note here that the parameters during the walking test were not influenced by the results of the EORTC QLQ-C30 questionnaire, QoR 15 questionnaire, and laboratory test data. We compared the subjects according to their level of physical activity before the surgery, and there was no difference. This may be because physical activity is self-assessed, which is a subjective measurement method. On the other hand, our respondents lacked planned, programmed, and structured physical activity (sport or recreation). Our respondents were primarily engaged in hobbies involving light-intensity physical activity. Those who rated their physical activity as moderate or high mostly referred to the self-assessment of their workplace.

Research from the 1980s consistently demonstrates that cardiac output and VO2 are increased postoperatively [34,35,36,37]. According to previous studies [35], there seems to be a common condition in which oxygen consumption during surgery is inadequate to meet intraoperative metabolic needs. This is thought to be due to decreased intraoperative cardiac output, leading to reduced oxygen delivery and altered intraoperative oxygen transport at the microcirculatory and cellular levels. The study above states that the altered physiological pattern during surgery is similar to that in a state of shock. For the purposes of this study, it is much more interesting to note that postoperative physiological changes were then considered to include increased cardiac output and increased oxygen delivery, which are necessary to meet the increased oxygen demand. These postoperative physiological changes may represent compensatory responses to intraoperative oxidative and metabolic deficits. It is also suggested that there are increased energy needs postoperatively due to the healing of the surgical wound. These increased needs contribute to the degree of metabolic increase required for recovery from surgery.

Acute physical (exercise or surgery) or psychological stressors result in the metabolic and hormonal changes that follow injury or trauma. They are part of a systemic response to injury encompassing various immunological, hematological, and endocrinological effects. After the study of the stress response to trauma, attention turned to the response to surgical trauma. The stress response to surgery is characterized by increased secretion of pituitary hormones and activation of the sympathetic nervous system. Changes in pituitary hormone secretion have secondary effects on hormone secretion from target organs. The overall metabolic effect of hormonal changes is increased catabolism that mobilizes substrates to provide energy sources and a mechanism for salt and water retention and maintaining fluid volume and cardiovascular homeostasis. [38,39]. This stimulates the cardiopulmonary, immune, musculoskeletal, and metabolic response [40]. The stress response and restoration of homeostasis increase metabolism, heart rate, and VO2 postoperatively.

Wearable devices supplement cancer care or cancer research with objective and reliable data on patients’ physical activity and add value by providing clinically relevant metrics that are otherwise difficult to capture. Garmin devices are also used in clinical settings for intervention and measurements. A search in a clinical trials database (https://clinicaltrials.gov/ (accessed on 30 June 2024)) revealed 41 studies using a Garmin wearable device [41]. In a systematic review of the use of wearable devices in oncology patients, the most commonly reported brand of wearable device was Garmin, with 13 studies.

We used the Garmin Vivosmart 4. The Garmin Vivosmart 4 is a wrist-worn activity tracker. It is based on an accelerometer that tracks steps, distance, heart rate, calories, floors climbed, and minutes of intense exercise. It also has a built-in display for direct control and data reading. The data can be analyzed later using the Garmin Connect mobile app version 5.13.0.27. Metric values were used to configure the device for the purposes of this study. The device was also configured to be worn on the left wrist and was placed proximal to the Caput ulnae sinistra for each subject, according to the manufacturer’s recommendations [42]. A study [42] was conducted to assess the validity of the Garmin Vivosmart device while walking at different speeds under controlled conditions. The results showed a tendency for the Garmin Vivosmart to count fewer steps compared to a manual pedometer. Despite these results, the activity tracker may still be clinically relevant at certain walking speeds. The study showed that the Garmin Vivosmart counted steps incorrectly at low and high speeds. We concluded that the Garmin Vivosmart is valid at 3.2 to 4.8 km/h speeds. Another paper discusses [43] how reliable such devices are at moderate walking speeds. A 2017 study reveals the validity of wearable devices in measuring heart rate compared to six-lead ECGs [44]. One of the wearable devices was also more accurate in measuring heart rate at rest and at lower exercise intensities. The Garmin Vivosmart HR was also found in the group of reliable devices at rest and lower exercise intensities. Since we used walking in our study (a very low-intensity exercise), we can conclude that there was no significant deviation from the wearable monitor. On the other hand, the same study confirmed that the chest-based wearable monitor had the smallest deviation from the six-lead ECG, and this may be a recommendation for using such a device in future studies. This device has not been validated on surgical or oncology patients, which is a significant limitation of this study.

The 6 min walk test may be recommended for use in cancer patients. The 6 min walk test is valid for objective and subjective measures and a repeatability test [25,26,45], but it is not valid for predicting peak VO2 in cancer patients [46]. The 6MWT has been used for over 50 years and has proven to be a useful screening tool for cardiorespiratory fitness in departments where cardiorespiratory exercise testing is unavailable [47]. The 6MWT does not provide incremental prognostic information for predicting moderate postoperative complications [48]. This test is also used in abdominal surgery to predict postoperative pulmonary complications [22,49]. The study shows that 6MWT is reliable, inexpensive, simple, and safe for predicting PPC, but measuring the VO2 peak is a better predictor than the 6MWT [50].

This study found a statistically significant difference between genders in walk distance, age in number of steps, and body composition in average and maximal heart rate during walking. A previous study reported a short-term decline in walking distance. In a study of patients over the age of 70 (76.0 ± 4.6) undergoing abdominal cancer surgery, a significant decline in physical performance was observed. The short-term walking distance reported in this study was a mean of 157 m (33%) [51]. A study including slightly younger patients with gastrointestinal cancer undergoing surgery (61.3 ± 11.0) reported a mean decline of 39.4 m (7%) during the 6MWT [52]. Another study reported a decline of 85 m (17%) in patients undergoing esophagectomy, mean age 67.3 ± 8.1, during the 6MWT [53]. One study using accelerometers allowed us to quantify physical activity in preoperative courses in colorectal surgery. The mean number of overall daily postoperative footsteps was significantly lower in participants with postoperative complications, and a significant correlation was found between the number of postoperative footsteps and length of stay in the hospital [54].

It is evident from the literature that the level of BMI, especially obesity [55], is positively related to the risk of colorectal cancer. This positive association is equally present in mice and women and unrelated to geographic location [56]. This connection is especially pronounced in younger adults [57]. Abdominal obesity can play an important role in the development of colorectal cancer [58,59]. Also, higher BMI hurt in-hospital mortality, but people with higher BMIs had significantly better long-term survival [60]. Also, obesity is associated with an increased risk of surgical site infections [61].

In our study, we had an increase in heart rate level after surgery in all three groups of patients according to BMI (normal weight, overweight, obese). When we divided the subjects into groups according to BMI, the results showed that obese people had a higher heart rate before and after surgery compared to people with normal body weight. On the other hand, people with excessive body weight had the same heart rate as people with normal weight before the surgery. Still, after the surgery, their heart rate increase was the highest, and, postoperatively, it approached the level of the obese.

Healthy, young, obese adults demonstrated higher cardiac index, cardiac output, and stroke volume during cycling exercise when comparing age, sex, and fitness level than normal-weight adults [62]. Also, cardiac output was increased at rest and during exercise in overweight young adults compared to normal-weight individuals. Neither group had arterial hypertension [63]. The aforementioned increase in heart function can be alleviated by regular exercise [64]. In obese people, an increase in the number of beats per minute was recorded during exercise but also in the rest phase after exercise [65]. Poor heart rhythm control is caused by increased sympathetic activity and decreased parasympathetic activity caused by obesity [66].

For the first time, our study describes the increase in heart rate after surgery with regard to body mass index. We included all types of colorectal cancer (colon, sigmoid, rectum, and rectosigmoid colon), as well as the open, traditional full-length approach and minimally invasive approach. We also included the duration of surgery, quality of life, quality of recovery, and laboratory findings. All the abovementioned parameters did not influence our subjects’ heart rate increases after surgery.

This study has several limitations. The 6 min walk test was validated and tested on cancer patients, colorectal cancer patients, and major abdominal surgery patients (most often in abdominal aorta surgery) but never on abdominal cancer surgery patients using the wearable device.

The major limitation of this study is that the Garmin Vivosmart 4 device used was an entry-level model, and there is very little evidence in the literature regarding its accuracy and precision in measuring heart rate and walking speed. According to previous research, there is the smallest deviation at rest and low exercise intensity. The relatively small sample size limits the generalization of our results. The study was conducted in patients undergoing cancer abdominal surgery, major abdominal surgery, and minimally invasive abdominal surgery. Larger studies are likely needed to develop models specific to various surgery types. On the other hand, we had a small number of laparoscopic surgery patients, and we could not estimate the impact of minimally invasive surgery on postoperative recovery. Most of the participants in our study did not engage in any kind of recreational activity or sports. Consequently, we do not have strong evidence to indicate a relationship between physical activity before surgery and better recovery after surgery. Another limitation referred to participants.

This result is directly applicable in clinical practice, especially for health professionals (physiotherapists) engaged in physical activity after surgery. The results of our study indicate changes in the 6MWT parameters before and after surgery. While we can observe a slower walk with a shorter distance and a smaller number of steps (in certain groups of subjects), information about the higher heart rate patients have after surgery is very useful. In particular, we are referring here to overweight and obese people. This information can help us properly plan and program physical activity after surgery.

What is particularly important is the planning and programming of the intervention in prehabilitation. Prehabilitation is defined as a process that includes the assessment of physical, nutritional, and psychological status to determine the baseline functional capacity, identify impairments, and plan an intervention to improve the preoperative functional reserve of the patient before the treatment itself [67]. Interventions address modifiable risk factors to improve treatment outcome (cancer) [68]. Prehabilitation is extremely important for oncology patients since they still have a whole series of treatment procedures to undergo after surgery. Therefore, it is very important that these patients recover from surgery as soon as possible and continue treatment as physically fit as possible. The data obtained in this study can be used to predict heart rate behavior before and after surgery and during prehabilitation. We can also see whether prehabilitation (physical preparation) reduces the difference in heart rate before and after surgery in the risk groups obtained in this study.

## 5. Conclusions

Taken together, to the best of our knowledge, this is the first study to compare data from a 6 min walk test before and after abdominal cancer surgery to describe patients’ specific physical recovery. Our study provides evidence that average and maximal heart rate during the 6 min walk test was higher during the postoperative period, especially in overweight and obese participants. Additionally, the study proved that the number of steps, walking distance, and average speed were lower post-surgery. We have no evidence that physical activity before surgery impacts these results.

Future studies should use ECGs or wearable chest devices to measure heart rate and perform the 6 min walk test on a treadmill. It is also important to include more laparoscopic surgeries or focus solely on open abdominal surgeries, excluding laparoscopic cases. Using bioimpedance instead of BMI provides a more accurate body composition measurement to assess post-surgery heart rate changes.

## Figures and Tables

**Figure 1 cancers-17-01782-f001:**
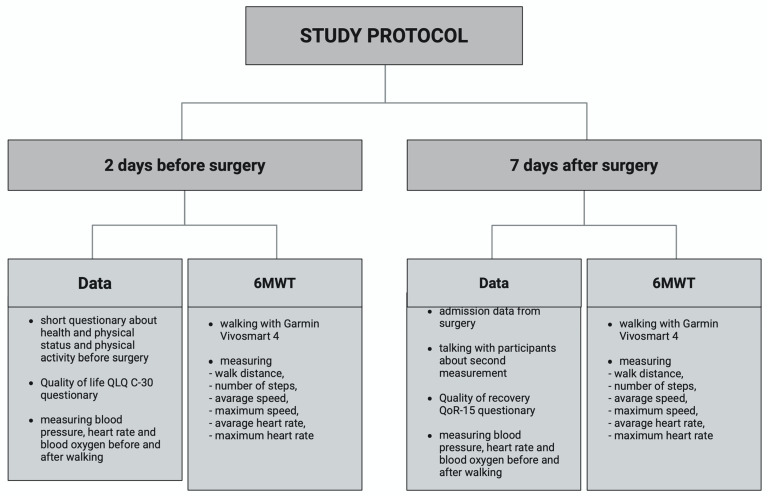
Research protocol overview.

**Figure 2 cancers-17-01782-f002:**
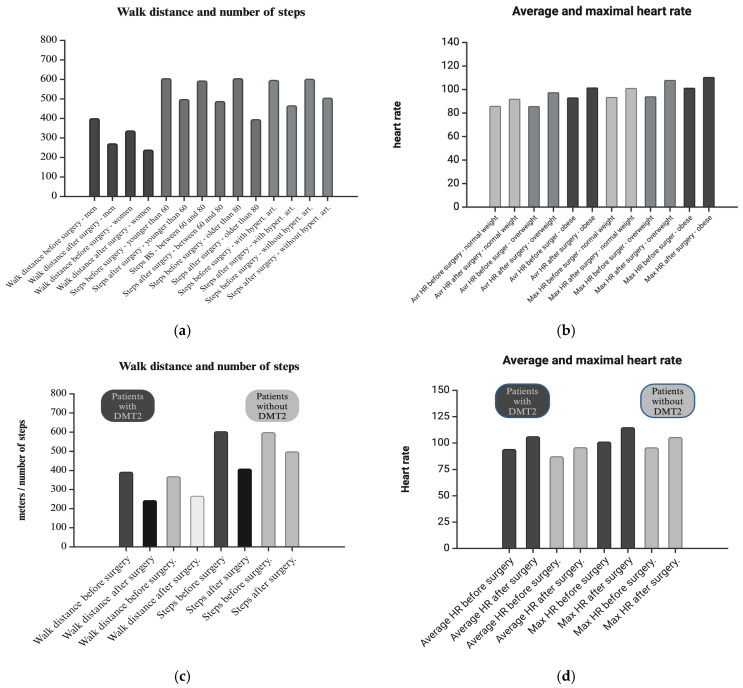
(**a**,**b**) Graphical overview of the difference between the groups for walk distance, number of steps, and heart rate. (**c**,**d**) Graphical overview difference between participants with and without Type 2 diabetes mellitus for walk distance, number of steps, and heart rate.

**Table 1 cancers-17-01782-t001:** Subjects’ besic characteristics.

N = 72			
		*n*	%
Gender	Male	47	65.3
	Female	25	34.7
Age	Younger than 60	26	36.11
	Between 60 and 75	40	55.5
	Older than 75	6	8.33
Localization of tumor	Right colon	17	23.6
	Left colon	24	33.3
	Rectum	31	43.1
Metastatic disease	No	55	76.4
	Yes	17	23.6
BMI	Underweight	0	0
	Normal	19	26.39
	Overweight	27	37.5
	Obese	26	36.11
Other diagnoses	Arterial hypertension	40	55.5
	Diabetes mellitus type 2	13	18.1
	Ischemic heart disease	7	9.7
	Other oncologic diagnosis	16	22.2
Cardiovascular drugs	Beta blocker	16	22.2
	Calcium channel blocker	13	1821
	ACE inhibitors	28	38.9
Weight loss	Yes	26	36.11
	No	46	63.89
Digestive symptoms	Diarrhea	12	16.67
	Nausea	2	2.78
	Constipation	10	13.89
	Bleeding	1	1.39
	No symptoms	47	65.3
Neoadjuvant therapy before surgery	No	60	83.3
	Chemotherapy/radiotherapy	8	11.11
	Chemotherapy	3	4.17
	Radiotherapy	1	1.39
Surgery procedure	Right colectomy	14	19.45
	Left colectomy	7	9.72
	Low anterior resection of rectum	14	19.45
	Segmental resection of colon	18	25
	Sec. Hartman	5	6.95
	Subtotal colectomy	3	4.17
	Abdominoperitoneal extirpation of the rectum	3	4.17
	Transanal	3	4.17
	Other	5	6.95
Duration of surgical procedure	Less than 2 h	26	36.11
	Between 2 and 3 h	35	48.61
	Longer than 3 h	11	15.27
Intensity of physical activity before surgery (workplace or free-time activity)	No physical activity (<1.5 METS)	5	6.94
	Light (<3 METs)	35	48.61
	Medium (z6METs)	22	30.55
	High (>1METs)	10	13.89

**Table 2 cancers-17-01782-t002:** Quality of life and quality of recovery.

	Global Health Status		Functional Scales					QoR 15	
			Physical	Role	Emotional	Cognitive	Social		
Mean	63.63		95.38	96.13	96.45	92.01	98.58	135.76	
SD	18.31		9.15	7.21	14.48	13.54	6.84	9.35	
**QLQ C-30 Before Surgery**									
	**Symptoms**								
	**Fatigue**	**Nausea and Vomiting**	**Pain**	**Dyspnea**	**Insomnia**	**Appetite Loss**	**Constipation**	**Diarrhea**	**Financial Difficulties**
Mean	5.94	1.64	5.4	5.32	10.65	5.33	11.54	9.68	1.05
SD	12.48	7.54	11.59	15.54	22.8	18.33	27.61	23.48	8.96

**Table 3 cancers-17-01782-t003:** Parameters evaluated during performance of 6MWT.

	Minimum	Maximum	Mean	Range	SD	*p*-Value
Maximal heart rate before surgery	57	121	96.85	64	13.01	*p* < 0.05
Maximal heart rate after surgery	77	151	107.3	74	13.44	
Average heart rate before surgery	51	114	88.75	63	12.66	*p* < 0.05
Average heart rate after surgery	61	125	97.79	64	12.12	
Maximal speed before surgery (km/h)	4.18	7.72	5.57	3.54	0.52	*p* < 0.05
Maximal speed after surgery (km/h)	2.42	7.89	5.15	5.47	0.71	
Average speed before surgery (km/h)	1.93	7.24	4.36	5.3	0.51	*p* < 0.05
Average speed after surgery (km/h)	0.48	6.43	3.46	5.95	0.7	
Number of steps before surgery	399	740	600.68	341	65.78	*p* < 0.05
Number of steps after surgery	100	651	485.37	551	130.61	
Walk distance before surgery (meters)	113	580	375.45	467	0.05	*p* < 0.05

## Data Availability

Data sharing not applicable. No new data were created or analyzed in this study.

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
