# Peer review of "Difference Between Walking Parameters During 6 Min Walk Test Before and After Abdominal Surgery in Colorectal Cancer Patients"

_cancers, 2025, doi:10.3390/cancers17111782_

Round 1

Reviewer 1 Report

Comments and Suggestions for Authors

Manuscript ID: Cancers-3616119

Title: Difference between walking parameters during 6-minute walk test before and after abdominal surgery in colorectal cancer patients

Summary: Colorectal cancer is a significant global health issue, with surgery being the primary treatment. This study evaluated pre-surgical physical fitness using cardiopulmonary exercise testing and the 6-minute walk test (6MWT) in patients awaiting open or laparoscopic surgery. Seventy-two patients participated, completing health questionnaires and performing the 6MWT before and after surgery. Results indicated significant differences in walking distance and heart rates based on gender and body mass index, highlighting the importance of assessing physical fitness before surgical interventions.

Please follow the suggestions below to improve the manuscript further. There are some corrections needed.

1. The authors. You don't need to write a summary in the paper if you have already written an abstract for a research article. Please remove the summary.

2. I recommend that the authors add a section at the end of the introduction briefly outlining the number of sections in the paper. This will help readers understand the structure and flow of the manuscript more clearly.

3. Please write clearly: the study design is a primary part of the authors' study.

4. The discussion section is too short; it’s insufficient to complete the study. Please write more and in detail.

5. How can the research be completed without conclusions and future research directions? Please pay more attention to writing them.

Author Response

Summary: Colorectal cancer is a significant global health issue, with surgery being the primary treatment. This study evaluated pre-surgical physical fitness using cardiopulmonary exercise testing and the 6-minute walk test (6MWT) in patients awaiting open or laparoscopic surgery. Seventy-two patients participated, completing health questionnaires and performing the 6MWT before and after surgery. Results indicated significant differences in walking distance and heart rates based on gender and body mass index, highlighting the importance of assessing physical fitness before surgical interventions.

Please follow the suggestions below to improve the manuscript further. There are some corrections needed.

Comment: The authors. You don't need to write a summary in the paper. If you have, Mary must be part of the manuscript. Thanks again.

Response: Thank you for pointing out the fact. I agree that a summary is not needed if we have an abstract. However, my mistake is that I did not write the full title (there was an error when editing the text; I did not paste the entire title text). Namely, this is a simple summary, which must be part of the manuscript according to the journal's rules, and I received it in a template. Also, according to the journal's rules, this paragraph must not have more than 400 characters. Therefore, I think that a simple summary. 

2. Comment: I recommend that the authors add a section at the end of the introduction, briefly outlining the number of sections in the paper. This will help readers understand the structure and flow of the manuscript more clearly.

Response: I will add the requested sentence.

3. Comment: Please write clearly: the study design is a primary part of the authors' study.

Response: Thanks for pointing out the omission,

Response: In the paragraph of exclusion criteria, I replaced the term "I can't exercise" with the specific conditions and diseases I meant by that term: orthopedic surgeries where recovery is not complete, lower limb amputations, degenerative or inflammatory diseases of the locomotor system that result in reduced mobility and difficulty walking (osteoarthritis, rheumatoid arthritis, ankylosing spondylitis, etc.), anemia where hemoglobin is below 80, cachexia, and conditions after a cerebrovascular accident where movement and walking are difficult.

The new paragraph of exclusion criteria reads: 

Excluded were patients with cardiorespiratory (uncontrolled hypertension, unstable angina pectoris, uncontrolled cardiac arrhythmia, or aortic aneurism more than 6,5 cm), musculoskeletal, and/or neurologic (impaired mobility, orthopedic surgeries where recovery is not complete, lower limb amputations, degenerative or inflammatory diseases of the locomotor system that result in reduced mobility and difficulty walking (osteoarthritis, rheumatoid arthritis, ankylosing spondylitis, etc.), anemia where hemoglobin is below 80, cachexia, and conditions after a cerebrovascular accident where movement and walking are difficult) conditions, those needing revision surgery, or with a second surgery within 6 months, psychiatric conditions or cognitive disorders, frail patients, and patients with severe anemia (hemoglobin less than 85).

  After the paragraph

Participants should rest for 15 minutes before the beginning of 6MWT. Before starting the test, blood pressure, heart rate, and oxygen saturation were measured. The same measurements were repeated after 6MWT. 

I add guidance for the 6-minute walk test

For the 6MWT, we used a 60-meter-long hospital corridor. Each subject was given clear instructions on how to perform the test. The aim of the test was to walk as far as possible. A physiotherapist accompanied the subject throughout the test, who noted how much time had passed every minute. The subject was also instructed that it was not advisable to talk during the test. If there were any questions or uncertainties, they should ask them before or after the test unless they had another problem with the 6MWT or the examiner asked them a question. They must say if they felt chest pain or dizziness during the test.

I explain the use of laboratory data.

For the purposes of this study, we also used some laboratory findings: hemoglobin and erythrocytes, which carry oxygen through the blood to muscle cells. Adequate oxygenation of muscle cells is especially important during physical activity, such as walking. On the other hand, we took inflammatory parameters that are routinely available to us (leukocytes and CRP) to determine whether inflammatory parameters affect reduced physical capacity after surgery.

I have added a diagram to show the flow of the research.

After paragraph

After admission to the hospital, participants who signed an informed consent form were included in a prospective cohort study. They completed a short questionnaire about health and physical status and preoperative physical activities. 

I add

After a short questionnaire, the subjects completed the EORTC QLQ-C30 quality of life questionnaire (European Organization for Research and Treatment of Cancer). We used the Quality of Recovery Questionnaire (QoR 15) 7 days after surgery. 

 I have supplemented the statistical analysis section.

Data were analyzed in the Statistica package, version 14.0.0 (TIBCO Data Science). Patient demographics, diagnosis, and treatment characteristics are presented as categorical variables using a frequency table. Continuous variables were presented with mean (standard deviation SD) or median (interquartile range IQR). For continuous variables, an independent samples t-test or the Mann-Whitney U test, as appropriate, was performed to analyze differences between the two measurements. 

We used ANOVA to analyze walking performance (number of steps, walking distance, average and maximal speed, average and maximal heart rate) during 6MWT. For the purposes of this test, certain adjustments to the data were necessary. It was necessary to divide the subjects into groups according to age, gender, diagnosis, other diagnoses, cardiovascular drug use, weight loss, digestive symptoms, treatment previous surgery, physical activity before surgery, BMI, type of surgery, and duration of surgery. We compared variables between the groups with ANOVA for dependent measurements. A multiple logistic regression analysis was performed to identify independent predictors of heart rate parameters during 6MWT using laboratory data, a quality of life questionnaire, and a quality of recovery questionnaire. The significance threshold was set at P value <0,05.

Comment: 4. The discussion section is too short; it's insufficient to complete the study. Please write more and in detail.

Response: 

Major surgery leads to higher metabolic demands, which are matched in healthy individuals by concomitant cardiac output and ventilation increases. We performed a prospective comparison study to compare walk distance, number of steps, speed, and heart rate during 6MWT before and after CRC surgery for different subgroups of patients.  

The data presented in this study show that postoperative patients had a higher maximum and average heart rate during the 6MWT. The average heart rate during walking was higher in overweight and obese people. It should be noted that people living with obesity had significantly higher heart rates before surgery. The situation was similar, with the maximum heart rate during the 6MWT. It is important to note here that people with arterial hypertension and ischemic heart diseases did not show a significant difference in average and maximum heart rate during walking before and after surgery. The reason may be the use of beta-blockers in pharmacologic therapy (all 16 participants with beta-blocker therapy were in the arterial hypertension group). Postoperatively, the average and maximum walking speed is lower, especially in patients with DMT2. All results are shown in Charts 1-4.

It is also important to note here that the parameters during the walking test were not influenced by the results of the EORTC QLQ-C30 questionnaire, QoR 15 questionnaire, and laboratory test data. We compared the subjects according to their level of physical activity before the surgery, and there was no difference. This may be because physical activity is self-assessed, which is a subjective measurement method. On the other hand, our respondents lack planned, programmed, and structured physical activity (sport or recreation). Our respondents are famously engaged in any hobby involving light-intensity physical activity. Those who rate their physical activity as moderate or high mostly refer to the self-assessment of their workplace.

Research from the 1980s consistently demonstrated that cardiac output and VO2 are increased postoperatively (34-37). According to previous studies (35), there seems to be a common condition in which oxygen consumption during surgery is inadequate to meet intraoperative metabolic needs. This is thought to be due to decreased intraoperative cardiac output, leading to reduced oxygen delivery and altered intraoperative oxygen transport at the microcirculatory and cellular levels. The study above states that the altered physiological pattern during surgery is similar to that in a state of shock. For the purposes of this study, it is much more interesting to note that postoperative physiological changes were then considered to include increased cardiac output and increased oxygen delivery, which are necessary to meet the increased oxygen demand. These postoperative physiological changes may represent compensatory responses to intraoperative oxidative and metabolic deficits. It is also suggested that there are increased energy needs postoperatively due to the healing of the surgical wound. These increased needs contribute to the degree of metabolic increase required for recovery from surgery.

Acute physical (exercise or surgery) or psychological stressors are given to the metabolic and hormonal changes that follow injury or trauma. It is part of a systemic response to injury encompassing various immunological, hematological, and endocrinological effects. After the study of the stress response to trauma, attention has turned to the response to surgical trauma. The stress response to surgery is characterized by increased secretion of pituitary hormones and activation of the sympathetic nervous system. Changes in pituitary hormone secretion have secondary effects on hormone secretion from target organs. The overall metabolic effect of hormonal changes is increased catabolism that mobilizes substrates to provide energy sources and a mechanism for salt and water retention and maintaining fluid volume and cardiovascular homeostasis. (38, 39). This stimulates the cardiopulmonary, immune, musculoskeletal, and metabolic response (40). The stress response and restoration of homeostasis increase metabolism, heart rate, and VO2 postoperatively. 

Wearable devices supplement cancer care or cancer research with objective and reliable data on patients' physical activity and add value by providing clinically relevant metrics that are otherwise difficult to capture. Garmin devices are also used in clinical settings for intervention and measurements. Conducting a search in a clinical trials database (clinicaltrails.gov) on 30. 6. 2024. revealed 41 studies using a Garmin wearable device (41). In a systematic review of the use of wearable devices in oncology patients, the most commonly reported brand of wearable devices was Garmin, with 13 studies. 

We used Garmin Vivosmart 4. The Garmin Vivosmart 4 is a wrist-worn activity tracker. It is based on an accelerometer that tracks steps, distance, heart rate, calories, floors climbed, and minutes of intense exercise. It also has a built-in display for direct control and data reading. The data can be analyzed later using the Garmin Connect mobile app. Metric values ​​were used to configure the device for the purposes of this study. The device was also configured to be worn on the left wrist and was placed proximal to the Caput ulnae sinistra for each subject, according to the manufacturer's recommendations. (42). A study (42) was conducted to assess the validity of the Garmin Vivosmart device while walking at different speeds under controlled conditions. The results showed a tendency for the Garmin Vivosmart to count fewer steps compared to a manual pedometer. Despite these results, the activity tracker may still be clinically relevant at certain walking speeds. The study showed that the Garmin Vivosmart counted steps incorrectly at low and high speeds. We conclude that the Garmin Vivosmart is valid at 3.2 to 4.8 km/h speeds. Another paper discusses (43) how reliable such devices are at moderate walking speeds. A 2017 study reveals the validity of wearable devices in measuring heart rate compared to six-lead ECG (44). One of the wearable devices was also more accurate in measuring heart rate at rest and at lower exercise intensities. The Garmin Vivosmart HR was also found in the group of reliable devices at rest and lower exercise intensities. Since we used walking in our study (a very low-intensity exercise), we can conclude that there was no significant deviation from the wearable monitor. On the other hand, the same study confirmed that the chest-based wearable monitor had the smallest deviation from the six-lead ECG, and this may be a recommendation for using such a device in future studies. This device has not been validated on surgical or oncology patients, which is a significant limitation of this study.

When we talk about the 6-minute walk test, It may be recommended for use in cancer patients. The 6-minute walk test is valid as a relationship for objective and subjective measures and a repeatability test (45-47), but it is not valid for predicting peak VO2 in cancer patients (48). The 6MWT has been used for over 50 years and has proven to be a useful screening tool for cardiorespiratory fitness in departments where cardiorespiratory exercise testing is unavailable (49). 6MWT did not provide incremental prognostic information for predicting moderate postoperative complications (50). This test is also used in abdominal surgery to predict postoperative pulmonary complications (51, 52). The study shows that 6MWT is s reliable, inexpensive, simple, and safe for predicting PPC, but measuring the VO2 peak is a better predictor than 6MWT (53). 

This study found a statistically significant difference between gender in walk distance, age in number of steps, and body composition in average and maximal heart rate during walking. A previous study reported a short-term decline in walking distance. In a study of patients over the age of 70 (76,0 +/-4,6) undergoing abdominal cancer surgery, a significant decline in physical performance was observed. The short-term walking distance reported in this study was a mean of 157 meters (33%) (54). A study including slightly younger patients with gastrointestinal cancer undergoing surgery (61,3 +/- 11,0) reported a mean decline of 39,4 meters (7%)  during 6MWT (55). Another study reported a decline in 85 meters (17%) in patients undergoing esophagectomy, mean age 67,3 +/- 8,1 during 6MWT (56). One study using accelerometers allowed us to quantify physical activity in preoperative courses in colorectal surgery. The mean number of overall daily postoperative footsteps was significantly lower in participants with postoperative complications, and a significant correlation was found between the number of postoperative footsteps and light of stay in the hospital (57). 

It is evident from the literature that the level of BMI, especially obesity (58), is positively related to the risk of colorectal cancer. This positive association is equally present in mice and women and unrelated to geographic location (59). This connection is especially pronounced in younger adults. (60). Abdominal obesity can play an important role in the development of colorectal cancer. (61, 62). Also, higher BMI hurt in-hospital mortality, but people with higher BMI had significantly better long-term survival (63). Also, obesity is associated with an increased risk of surgical site infections (64). 

In our study, we had an increase in heart rate level after surgery in all three groups of patients according to BMI (normal weight, overweight, obese). When we divided the subjects into groups according to BMI, the results showed that obese people had a higher heart rate before and after surgery compared to people with normal body weight. On the other hand, people with excessive body weight had the same heart rate as people with normal weight before the operation. Still, after the operation, their heart rate increase was the highest, and postoperatively, it was approached at the level of the obese. 

Healthy young obese adults demonstrated higher cardiac index, cardiac output, and stroke volume during cycling exercise compared to age, sex, and fitness level than normal-weight adults (65). Also, cardiac output was increased at rest and during exercise in overweight young adults compared to normal-weight individuals. Both groups didn't have arterial hypertension (66). The aforementioned increase in heart function can be alleviated by regular exercise (67). In obese people, an increase in the number of beats per minute was recorded during exercise but also in the rest phase after exercise (68). Poor heart rhythm control is caused by increased sympathetic activity and decreased parasympathetic activity caused by obesity (69). 

For the first time, our study describes the increase in heart rate after surgery with regard to body mass index. We included all types of colorectal cancer (colon, sigmoid, rectum, and rectosigmoid colon) as well as open traditional full-length approach and minimally invasive approach. We also included the duration of surgery, quality of life, quality of recovery, and laboratory findings. All the abovementioned parameters did not influence our subjects' heart rate increase after surgery.

This study has several limitations. The 6-minute walk test was validated and tested on cancer patients,  colorectal cancer patients, and major abdominal surgery patients (most often in abdominal aorta surgery) but never in abdominal cancer surgery using the wearable device. 

The major limitation of this study is that the Garmin Vivosmart 4 device used was an entry-level model, and there is very little evidence in the literature regarding its accuracy and precision in measuring heart rate and walking speed. According to previous research, there is the smallest deviation at rest and low exercise intensity. The relatively small sample size limits the generalization of our results. The study was conducted in patients undergoing cancer abdominal surgery, major abdominal surgery, and minimally invasive abdominal surgery. Larger studies are likely needed to develop models specific to various surgery types. On the other side, we had a small number of laparoscopic surgery, and we could not estimate the impact of minimally invasive surgery on postoperative recovery. Most of the participants in our study did not engage in any kind of recreational activity or sports. Consequently, we do not have strong evidence to indicate a relationship between physical activity before surgery and better recovery after surgery. The biggest limitation of this study is that the device Garmin Vivosmart 4 is used as an entry-level model, and there is no literature on its accuracy and precision in measuring heart rate and walking speed. Another limitation referred to participants. 

This result is directly applicable in clinical practice, especially for health professionals (physiotherapists) engaged in physical activity after surgery. The results of our study indicate changes in 6MWT parameters before and after surgery. While we can observe a slower walk with a shorter distance and a smaller number of steps (in certain groups of subjects), information about the higher heart rate patients have after surgery is very useful. In particular, we are referring here to overweight and obese people. This information can help us properly plan and program physical activity after surgery.

This result is directly applicable in clinical practice, especially for healthcare professionals (physiotherapists) involved in physical activity after surgery. The results of our study indicate changes in 6MWT parameters before and after surgery. While we may observe a slower gait with a shorter perceived distance and fewer steps (in certain groups of subjects), information about patients' higher heart rate after surgery is very useful. Here, we are thinking especially of overweight and obese people. This information can help us properly plan and program physical activity after surgery.

What is particularly important is the planning and programming of the intervention in prehabilitation. Prehabilitation is defined as a process that includes the assessment of physical, nutritional, and psychological status to determine the baseline functional capacity, identify impairments, and plan an intervention to improve the preoperative functional reserve of the patient before the treatment itself (70). Interventions address modifiable risk factors to improve treatment outcome (cancer) (71). Prehabilitation is extremely important for oncology patients since they still have a whole series of treatment procedures to undergo after surgery. Therefore, it is very important that these patients recover from surgery as soon as possible and continue treatment as physically fit as possible. The data obtained in this study can be used to predict heart rate behavior before and after surgery and during prehabilitation. We can also see whether prehabilitation (physical preparation) reduces the difference in heart rate before and after surgery in the risk groups obtained in this study.

5. How can the research be completed without conclusions and future research directions? Please pay more attention to writing them.

Conclusion

Taken together, to the best of our knowledge, this is the first study to compare data from a 6-minute walk test before and after abdominal cancer surgery to describe patients' specific physical recovery. Our study provides evidence that average and maximal heart rate during the 6-minute walk test was higher during the postoperative period, especially in overweight and obese participants. Additionally, the study proved that the number of steps, walking distance, and average speed were lower post-surgery. We have no evidence that physical activity before surgery impacts these results. 

Future studies should use ECG or wearable chest devices to measure heart rate and perform the 6-minute walk test on a treadmill. It is also important to include more laparoscopic surgeries or focus solely on open abdominal surgeries, excluding laparoscopic cases. Using bioimpedance instead of BMI provides a more accurate body composition measurement to assess post-surgery heart rate changes.

Reviewer 2 Report

Comments and Suggestions for Authors

The manuscript presents a prospective cohort study evaluating pre- and post-operative physical performance in colorectal cancer (CRC) patients using the 6-minute walk test (6MWT). The use of wearable technology (Garmin Vivosmart 4) to track walking distance, heart rate, and step count introduces a modern, practical approach to perioperative physical assessment. While the topic is clinically relevant and the dataset is respectable (n = 72), the manuscript currently falls short in multiple areas: methodological clarity, statistical robustness, and language quality. Furthermore, the manuscript offers limited novelty, as similar studies exist, and it does not provide significant advancement in clinical management or decision-making tools.I therefore recommend major revisions, with careful reconsideration of the study's added value and improvements in structure, discussion depth, and interpretation of results.

Major Comments

1. Limited Novelty and Contribution. Several previous studies have evaluated changes in functional capacity before and after major abdominal or oncologic surgery using 6MWT or other performance-based measures. The authors should clarify what this study adds, beyond confirming well-known findings (e.g., reduced walking capacity post-surgery, obesity-related higher HR). The use of a consumer-grade wearable alone is not sufficient novelty. A more focused discussion on how these findings could inform prehabilitation or postoperative rehab strategies is needed.

2. Methodological Concerns. Inclusion/exclusion criteria are only partially clear. For example, were all stages of CRC included? Were emergency surgeries excluded? There is heterogeneity in surgical types and approaches, but subgroup analysis by surgical technique (open vs. laparoscopic) is not performed. The justification for 6MWT timing (2 days before, 7 days after) is not well explained. Clearly define the clinical homogeneity of the cohort. Consider whether any subgroup analysis (e.g., by surgical approach) could be performed to reduce bias.

3. Statistical Interpretation Needs Caution. The multiple comparisons (age, gender, BMI, comorbidities) increase the risk of Type I error. No correction (e.g., Bonferroni) was applied for multiple testing. The regression model did not yield significant predictors, yet the discussion implies causality in several areas. Acknowledge the risk of overinterpretation of weak associations. Consider adjusting for multiple comparisons.

4. Device Validity and Measurement Accuracy The Garmin Vivosmart 4 is not a medical-grade device. While some validation is discussed, its use in postoperative surgical patients is not validated. Emphasize this limitation more strongly in the discussion. Suggest validation against standard methods (e.g., HR monitors with chest straps) in future studies.

5. Language and Structure Require Improvement. The manuscript contains frequent grammar errors, awkward phrasing, and non-standard scientific terminology (e.g., “visibled”, “participants tho cannot exercise”). A thorough English language revision by a professional is necessary before publication.

Minor Comments

  • The figures and tables are helpful but need consistent formatting and legends.

  • Clarify abbreviations (e.g., DMT2) at first use.

  • Improve the abstract: mention study design, sample size, and key findings with statistical significance.

Author Response

The manuscript presents a prospective cohort study evaluating pre- and postoperative physical performance in colorectal cancer (CRC) patients using the 6-minute walk test (6MWT). The use of wearable technology (Garmin Vivosmart 4) to track walking distance, heart rate, and step count introduces a modern, practical approach to perioperative physical assessment. While the topic is clinically relevant and the dataset is respectable (n = 72), the manuscript currently falls short in multiple areas: methodological clarity, statistical robustness, and language quality. Furthermore, the manuscript offers limited novelty, as similar studies do not advance clinical management or decision-making tools significantly. Therefore, I recommend major revisions, carefully reconsidering the study's added value and improvements in structure, discussion depth, and interpretation of results.

Major Comments

Comment: Limited Novelty and Contribution. Several previous studies have evaluated changes in functional capacity before and after major abdominal or oncologic surgery using 6MWT or other performance-based measures. The authors should clarify what this study adds beyond confirming well-known findings (e.g., reduced walking capacity post-surgery, obesity-related higher HR). The use of a consumer-grade wearable alone is not a sufficient novelty. A more focused discussion is needed on how these findings could inform prehabilitation or postoperative rehab strategies.

Response: I agree with your comment. To that end, I have supplemented the part of the discussion that discusses the results' applicability in clinical practice. The text that I expanded on is below.

This result is directly applicable in clinical practice, especially for healthcare professionals (physiotherapists) involved in physical activity after surgery. The results of our study indicate changes in 6MWT parameters before and after surgery. While we may observe a slower gait with a shorter perceived distance and fewer steps (in certain groups of subjects), information about the higher heart rate that patients have after surgery is beneficial. Here, we are thinking primarily of overweight and obese people. This information can help us properly plan and program physical activity after surgery.

What is particularly important is the planning and programming of the intervention in prehabilitation. Prehabilitation is defined as a process that includes the assessment of physical, nutritional, and psychological status to determine the baseline functional capacity, identify impairments, and plan an intervention to improve the preoperative functional reserve of the patient before the treatment itself (70). Interventions address modifiable risk factors to improve treatment outcomes (cancer) (71). Prehabilitation is extremely important for oncology patients since they still have a whole series of treatment procedures after surgery. Therefore, these patients must recover from surgery as soon as possible and continue treatment as physically fit as possible. The data obtained in this study can be used to predict heart rate behavior before and after surgery and during rehabilitation. We can also see whether prehabilitation (physical preparation) reduces the difference in heart rate before and after surgery in the risk groups obtained in this study.

2. Comment: Methodological Concerns. Inclusion/exclusion criteria are only partially clear. For example, were all stages of CRC included? Were emergency surgeries excluded? There is heterogeneity in surgical types and approaches, but subgroup analysis by surgical technique (open vs. laparoscopic) is not performed. The justification for 6MWT timing (2 days before, 7 days after) is poorly explained. Clearly define the clinical homogeneity of the cohort. Consider whether any subgroup analysis (e.g., by surgical approach) could be performed to reduce bias.

Response: At the Clinic for tumors, Sestre milosrdnice UHC, Zagreb, Croatia, where I conducted the research, there are no emergency surgeries, so I did not list them in the exclusion criteria, stage IV cancer is most often accompanied by poor mobility, poor general condition and/or anemia. Therefore, these patients were excluded according to other criteria, but I will also add stage IV cancer to the exclusion criteria.

The amended paragraph of the exclusion criteria reads:

Excluded were patients with cardiorespiratory (uncontrolled hypertension, unstable angina pectoris, uncontrolled cardiac arrhythmia, or aortic aneurism more than 6,5 cm), musculoskeletal, and/or neurologic (impaired mobility, participants who cannot exercise) conditions, those needing revision surgery, or with a second surgery within 6 months, emergency surgery, stage IV of cancer, psychiatric conditions or cognitive disorders, frail patients, and patients with severe anemia (hemoglobin less than 85).

We included patients in the study after hospitalization, usually 2 days before the procedure. I did not want to include them the day before the surgery because that is when the preparation for the surgery begins (premedication and bowel cleansing). Seven days after the surgery is approximately before discharge home from the hospital since patients at the Clinic are, on average, 8-10 days postoperative. 

In the section on study protocol, I add a new paragraph: 

All participants listed for CRC surgery, as per standard care, are evaluated by the hospital's multidisciplinary team (surgeon, anaesthesiologist, and nurse). They received nutritional support pre- or post-operatively, where needed, provided by a nurse/stomal therapist. Also, all participants were mobilized the day after surgery. 

We included patients in the study after hospitalization, which is 2 days after surgery. We performed the postoperative measurement 7 days after surgery, approximately 1-3 days before discharge from the hospital.

After admission to the hospital, participants who signed an informed consent form were included in a prospective cohort study. They completed a short questionnaire about health, physical status, and preoperative physical activities. 

I conducted a subgroup analysis, considering the type of surgery (see the section on differences between groups) in the results. No significant statistical difference was obtained between the two groups. The reason for this may be the minimal number of laparoscopic surgeries, which I will mention in the discussion as one of the limitations of the research. 

Section results  - the difference between group

We compared measured values during 6MWT using by group. Grouping variables used included gender (men and women), age (younger than 60, between 60 and 80, and older than 80 years), localization of tumor (rectum, sygmoid colon, right hemicolon, left hemicolon, colorectal), other diseases (arterial hypertension, heart disease, type 2 diabetes, other oncological diseases), cardiovascular drugs (beta blockers ACE inhibitors, angiotensin II receptor blockers), treatment previous surgery (chemotherapy/radiotherapy, chemotherapy or radiotherapy) level of physical activity before surgery (low, medium or high level), BMI normal weight, overweight, obese), type of surgery (major open abdomilal or laparoscopic) and duration of surgery (up to 2 hours, between 2 and 3 hours more than 3 hours). 

3. Comment: Statistical Interpretation Needs Caution. The multiple comparisons (age, gender, BMI, comorbidities) increase the risk of Type I error. No correction (e.g., Bonferroni) was applied for multiple testing. The regression model did not yield significant predictors, yet the discussion implies causality in several areas. Acknowledge the risk of overinterpretation of weak associations. Consider adjusting for multiple comparisons.

Response: Thank you for pointing out my mistake. I used Tukey to correct the results, but unfortunately, I forgot to mention that detail when I wrote the statistical analysis section. I will add the correction. Thank you.

The supplemented paragraph of the statistical analysis reads:

Data were analyzed in the Statistica package, version 14.0.0. (TIBCO Data Science). Patient demographics, diagnosis, and treatment characteristics are presented as categorical variables. These variables are presented using a frequency table. Continuous variables were presented with mean (standard deviation SD) or median (interquartile range IQR). For continuous variables, an independent samples t-test or the Mann-Whitney U test, as appropriate, was performed to analyze differences between the two measurements. 

For the analysis of walking performance (number of steps, walking distance, average and maximal speed, average and maximal heart rate) during 6MWT, we used ANOVA. For this test, specific adjustments to the data were necessary. It was essential to divide the subjects into groups according to age, gender, diagnosis, other diagnoses, cardiovascular drug used, weight loss, digestive symptoms, treatment privius surgery, physical activity before surgery, BMI, type of surgery, and duration of surgery. We compared variables between the groups with ANOVA for dependent measurements. We used Tukey SSD for data correction. A multiple logistic regression analysis was performed to identify independent predictors of heart rate parameters during 6MWT using laboratory data, a quality of life questionnaire, and a quality of recovery questionnaire. The significance threshold was set at P value <0,05. 

4. Comment: Device Validity and Measurement Accuracy The Garmin Vivosmart 4 is not a medical-grade device. While some validation is discussed, its use in postoperative surgical patients is not validated. Emphasize this limitation more strongly in the discussion. Suggest validation against standard methods (e.g., HR monitors with chest straps) in future studies.

Response: Yes, I agree that the text referring to the wearable device was too short. I expanded the passage in the discussion referring to the Garmin Vivosmart 4 and the entire passage is below.

We used Garmin Vivosmart 4. The Garmin Vivosmart 4 is a wrist-worn activity tracker. It is based on an accelerometer that tracks steps, distance, heart rate, calories, floors climbed, and minutes of intense exercise. It also has a built-in display for direct control and data reading. The data can be analyzed later using the Garmin Connect mobile app. Metric values ​​were used to configure the device for the purposes of this study. The device was also configured to be worn on the left wrist and was placed proximal to the Caput ulnae sinistra for each subject, according to the manufacturer's recommendations. (42). A study (42) was conducted to assess the validity of the Garmin Vivosmart device during walking at different walking speeds under controlled conditions. The results showed a tendency for the Garmin Vivosmart to count fewer steps compared to a manual pedometer. Despite these results, the activity tracker may still be clinically relevant at certain walking speeds. The results of the study showed that the Garmin Vivosmart miscounted steps at both low and high speeds. We conclude that the Garmin Vivosmart is valid at speeds of 3.2 to 4.8 km8h. Another paper discusses (43) how reliable such devices are at moderate walking speeds. A 2017 study reveals the validity of wearable devices in measuring heart rate compared to six-lead ECG (44). It was also found that one of the wearable devices was more accurate in measuring heart rate at rest and lower exercise intensities. The Garmin Vivosmart HR was also found in the group of devices that are reliable at rest and lower exercise intensities. Since we used walking in our study (a very low-intensity exercise), we can conclude that there was no significant deviation from the wearable monitor. On the other hand, the same study confirmed that the chest-based wearable monitor had the slightest deviation from the six-lead ECG, and this may be a recommendation for using such a device in future studies. This device has not been validated on surgical or oncology patients, which is a significant limitation of this study.

5. Comment:  Language and Structure Require Improvement. The manuscript contains frequent grammar errors, awkward phrasing, and non-standard scientific terminology (e.g., "visible," "participants who cannot exercise"). A thorough English language revision by a professional is necessary before publication.

Response: Yes, I agree with you. It is necessary to review the entire text of the articles and correct any grammatical and typographical errors that may have been made.

Minor Comments

The figures and tables are helpful but need consistent formatting and legends.

Clarify abbreviations (e.g., DMT2) at first use.

Improve the abstract: mention study design, sample size, and key findings with statistical significance.

Reviewer 3 Report

Comments and Suggestions for Authors

Thank you for the opportunity to review this article. This is well written article. I have only minor comments about methods.

The exclusion criteria are extensive but presented in a dense and at times unclear manner. For example, the phrase "participants who cannot exercise" should be revised for clarity.

While the timing and device used for the 6MWT are described, further details are needed to ensure reproducibility. For instance, was the walking course standardized (e.g., 30-meter hallway), and were verbal encouragements given according to established guidelines?

The rationale for selecting specific laboratory parameters (e.g., CRP, hemoglobin) and their hypothesized link to postoperative walking performance should be briefly explained.

Clarify when exactly the QLQ-C30 and QoR-15 questionnaires were administered in relation to the surgery. Were both administered preoperatively, or was QoR-15 also used postoperatively?

The statistical analysis section would benefit from more detail regarding how missing data were handled and whether any adjustments were made for multiple comparisons.

Some minor grammatical errors (e.g., “participants who cannot exercise,” “Garmin td.”) need correction to ensure professional and accurate presentation.

Comments on the Quality of English Language

some sophistication are needed

Author Response

Comment: The exclusion criteria are extensive but presented in a dense and, at times, unclear manner. For example, the phrase "participants who cannot exercise" should be revised for clarity.

Response: Thanks for pointing out the omission,

In a paragraph of exclusion criteria, I replaced the term "I can't exercise" with the specific conditions and diseases I meant by that term: orthopedic surgeries where recovery is not complete, lower limb amputations, degenerative or inflammatory diseases of the locomotor system that result in reduced mobility and difficulty walking (osteoarthritis, rheumatoid arthritis, ankylosing spondylitis, etc.), anemia where hemoglobin is below 80, cachexia, and conditions after a cerebrovascular accident where movement and walking are difficult.

The new paragraph of exclusion criteria reads: 

Excluded were patients with cardiorespiratory (uncontrolled hypertension, unstable angina pectoris, uncontrolled cardiac arrhythmia, or aortic aneurism more than 6,5 cm), musculoskeletal, and/or neurologic (impaired mobility, orthopedic surgeries where recovery is not complete, lower limb amputations, degenerative or inflammatory diseases of the locomotor system that result in reduced mobility and difficulty walking (osteoarthritis, rheumatoid arthritis, ankylosing spondylitis, etc.), anemia where hemoglobin is below 80, cachexia, and conditions after a cerebrovascular accident where movement and walking are difficult) conditions, those needing revision surgery, or with a second surgery within 6 months, psychiatric conditions or cognitive disorders, frail patients, and patients with severe anemia (hemoglobin less than 85).

Comment:  While the timing and device used for the 6MWT are described, further details are needed to ensure reproducibility. For instance, was the walking course standardized (e.g., 30-meter hallway), and were verbal encouragements given according to established guidelines?

Response: After the paragraph

Participants should rest for 15 minutes before the beginning of 6MWT. Before starting the test, blood pressure, heart rate, and oxygen saturation were measured. The same measurements were repeated after 6MWT. 

I add guidance for the 6-minute walk test.

For the 6MWT, we used a 60-meter-long hospital corridor. Each subject was given clear instructions on how to perform the test. The aim of the test was to walk as far as possible. The subject was accompanied by a physiotherapist throughout the test, who noted how much time had passed every minute. The subject was also instructed that it was not advisable to talk during the test. If there were any questions or uncertainties, they should ask them before or after the test unless they had another problem with the 6MWT or the examiner asked them a question. They must say if they felt chest pain or dizziness during the test.

Comment: The rationale for selecting specific laboratory parameters (e.g., CRP, hemoglobin) and their hypothesized link to postoperative walking performance should be briefly explained.

Respornse: thanks for your comment. For the purposes of this study, we also used some laboratory findings. Hemoglobin and erythrocytes that carry oxygen through the blood to muscle cells. Adequate oxygenation of muscle cells is especially important during physical activity, in this case during walking. On the other hand, we took inflammatory parameters that are routinely available to us (leukocytes and CRP) to determine whether inflammatory parameters affect reduced physical capacity after surgery.

Comment: Clarify when exactly the QLQ-C30 and QoR-15 questionnaires were administered about the surgery. Were both administered preoperatively or was QoR-15 also used postoperatively?

Response: Thank you for pointing out the shortcoming, I completely agree with you and I have added an explanation of when these two questionnaires were used and I have added a diagram to show the flow of the research.

After paragraph

After admission to the hospital participants who signed an informed consent were included in a prospective cohort study. They fulfilled a short questionnaire about health and physical status, and preoperative physical activities. 

I add

After a short questionnaire, the subjects completed the EORTC QLQ-C30 quality of life questionnaire (European Organization for Research and Treatment of Cancer). We used the Quality of Recovery Questionnaire (QoR 15) 7 days after surgery.

Comment: The statistical analysis section would benefit from more detail regarding how missing data were handled and whether any adjustments were made for multiple comparisons.

Response: Thank you. I have supplemented the statistical analysis section.

Data were analyzed in the Statistica package, version 14.0.0. (TIBCO Data Science). Patient demographics, diagnosis, and treatment characteristics are presented as categorical variables. These variables are presented using a frequency table. Continuous variables were presented with mean (standard deviation SD), or median (interquartile range IQR). For continuous variables, an independent samples t-test or the Mann-Whitney U test, as appropriate, was performed to analyze differences between the two measurements. 

For the analysis of walking performance (number of steps, walking distance, average and maximal speed, average and maximal heart rate) during 6MWT, we used ANOVA.s For the purposes of this test, specific adjustments to the data were necessary. It was necessary to divide the subjects into groups according to age, gender, diagnosis, other diagnoses, cardiovascular drug use, weight loss, digestive symptoms, treatment previous surgery, physical activity before surgery, BMI, type of surgery, and duration of surgery. We compared variables between the group with ANOVA for dependent measurements. A multiple logistic regression analysis was performed to identify independent predictors of heart rate parameters during 6MWT using laboratory data, a quality of life questionnaire, and a quality of recovery questionnaire. The significance threshold was set at P value <0,05. 

Some minor grammatical errors (e.g., "participants who cannot exercise," "Garmin td.") need correction to ensure professional and accurate p

Yes, I completely agree with you. It is necessary to review the entire text of the articles and correct any grammatical and typographical errors that may have been made.

Round 2

Reviewer 1 Report

Comments and Suggestions for Authors

Thank you for your thorough revisions; the paper is now ready for publication.

Reviewer 2 Report

Comments and Suggestions for Authors

The authors adequately responded to all the comments.